IPPP/25/18, P3H-25-023, KA-TP-09-2025, MCNET-25-05, ZU-TH 20/25

# EERAD3 version 2:
# QCD corrections in hadronic colour-singlet decays

**Benjamin Campillo Aveleira[1⋆] Aude Gehrmann-De Ridder[2,3†], Thomas Gehrmann[3‡], Nigel Glover[4•], Gudrun Heinrich[1∗], and Christian T. Preuss[5,6+]**

**1** Institute for Theoretical Physics, KIT, 76131 Karlsruhe, Germany
**2** Institute for Theoretical Physics, ETH, 8093 Zürich, Switzerland
**3** Department of Physics, University of Zürich, 8057 Zürich, Switzerland
**4** Institute for Particle Physics Phenomenology, Department of Physics, Durham University, DH1 3LE, U.K.
**5** Department of Physics, University of Wuppertal, 42119 Wuppertal, Germany
**6** Institut für Theoretische Physik, Georg-August-Universität Göttingen, 37077 Göttingen, Germany

⋆ benjamin.campillo@kit.edu , † gehra@phys.ethz.ch , ‡ thomas.gehrmann@uzh.ch ,
• e.w.n.glover@durham.ac.uk , ∗ gudrun.heinrich@kit.edu ,
+ christian.preuss@uni-goettingen.de

## Abstract

We present a major update of the publicly available EERAD3 package to calculate perturbative corrections in the strong coupling in hadronic Higgs and $Z$-boson decays. We describe the theoretical framework underlying the numerical implementation and provide a guide to the usage of the program.

# 1 Introduction

The study of jets and event shapes in $e^+e^-$ annihilation has been vital to advance perturbative QCD, and led to legacy determinations of the strong coupling $\alpha_s$ [1–7]. Today, event shape observables are still of major importance, for example to study non-perturbative effects that will become important at the precision that can be reached by future high energy lepton colliders. Such colliders, also called "Higgs factories", will furthermore provide unique opportunities to study Higgs boson properties through hadronic Higgs decays, in particular for channels which are inaccessible at the LHC due to large backgrounds, such as $H \to gg$ or $H \to c\bar{c}$, or only accessible in combination with tagging particles, such as $H \to b\bar{b}$ through $pp \to VH$.

Event shapes in $e^+e^-$ annihilation related to three-jet production at NNLO first have been presented in Refs. [8–10], NNLO moments of event shapes can be found in Refs. [11, 12], and NNLO distributions for event-orientation observables in $e^+e^-$ annihilation to three jets have been presented in [13]. Three-jet rates at NNLO have been calculated in Refs. [14, 15] based on antenna subtraction [16, 17] and the CoLoRFulNNLO subtraction method [18, 19] has been applied to calculate jet rates and event shapes in Refs. [20–22]. Soft-drop event shapes have been considered using the CoLoRFulNNLO method in [23]. The combination of fixed order NNLO results for event shapes with resummation in the infrared-sensitive regions has proven very beneficial, not only for determinations of $\alpha_s$, but also for studies of non-perturbative effects manifesting themselves as power corrections, see e.g. Refs. [3, 24–29] for recent developments. Four-jet event shapes in $e^+e^-$ annihilation have been calculated at NLO in [30–38].

Event shape distributions for hadronic Higgs boson decays to three jets at NLO, originating either from Higgs decays to a bottom-quark pair or from a loop-induced decay to two gluons in the heavy-top limit (HTL) have been calculated in Ref. [39] and the matching to NLL resummation has been performed in Ref. [40]. The framework of [39, 40] has also been extended to include infrared-safe flavoured jet algorithms and flavour-sensitive observables in [41]. An-

gularities in hadronic Higgs decays have been studied in [42, 43]. Four-jet event shapes in hadronic Higgs decays been presented at NLO in Ref. [44]. Recently, jet rates in the Durham algorithm have been calculated to third order in hadronic Higgs decays in [45].

Here we present a program package that can provide the above-mentioned processes in one framework: 3-jet production (and related event shapes) in $e^+e^-$ annihilation at NNLO QCD, 4-jet production in $e^+e^-$ annihilation at NLO QCD, as well as 3-jet and 4-jet production in $H \to b\bar{b}$ and $H \to gg$ at NLO. It is an upgrade and major extension of the program EERAD3, which was originally published in Ref. [46]. Apart from the larger number of available processes, the program has undergone a major technical overhaul, leading to more efficient phase space sampling and a python-based user interface. The implementation of flavour-sensitive observables as described in [41] is planned for a subsequent release.

The structure of this work is oriented at a program description for the practitioner. In section 2 we briefly describe the theoretical framework, in section 3 we comment on the available processes. Section 4 is dedicated to a description of the program structure, installation and usage are described in section 5, before we summarise in section 6. In the Appendix we give example runcards for all six available processes.

## 2   Theoretical framework

The EERAD3 Monte-Carlo generator calculates QCD corrections to infrared-safe event-shape observables $y$ in hadronic colour-singlet decays. Denoting the invariant mass of the decaying resonance by $\sqrt{s}$, the perturbative expansion of event-shape distribution can be written at scale $\mu^2 = s$ as

$$\frac{1}{\Gamma_{jj}^{(n)}(s)} \frac{d\Gamma}{dy}(s) = \left(\frac{\alpha_s(\sqrt{s})}{2\pi}\right) \frac{d\bar{A}}{dy} + \left(\frac{\alpha_s(\sqrt{s})}{2\pi}\right)^2 \frac{d\bar{B}}{dy} + \left(\frac{\alpha_s(\sqrt{s})}{2\pi}\right)^3 \frac{d\bar{C}}{dy} + \mathcal{O}(\alpha_s^4). \tag{1}$$

Here, the event-shape distribution has been normalised to the inclusive two-parton decay width $\Gamma_{jj}^{(n)}$, where $n$ denotes the order of the calculation, with $n = 0$ referring to LO, $n = 1$ to NLO, and $n = 2$ to NNLO. Because the perturbative coefficients $\bar{A}$, $\bar{B}$, and $\bar{C}$ are calculated at fixed renormalisation scale $\mu^2 = s$, they depend only on the value of the observable $y$.

To account for the running of the strong coupling constant when evaluating eq. (1) at a scale $\mu^2 \neq s$, the renormalisation-group equation of the strong coupling up to third order,

$$\mu^2 \frac{d\alpha_s(\mu)}{d\mu^2} = -\alpha_s(\mu) \left(\beta_0 \left(\frac{\alpha_s(\mu)}{2\pi}\right) + \beta_1 \left(\frac{\alpha_s(\mu)}{2\pi}\right)^2 + \beta_2 \left(\frac{\alpha_s(\mu)}{2\pi}\right)^3 + \mathcal{O}(\alpha_s^4)\right), \tag{2}$$

is solved by introducing an integration constant $\Lambda$ with $L = \log(\mu^2/\Lambda^2)$ to yield

$$\alpha_s(\mu) = \frac{2\pi}{\beta_0 L} \left(1 - \frac{\beta_1}{\beta_0^2} \frac{\log L}{L} + \frac{1}{\beta_0^2 L^2} \left(\frac{\beta_1^2}{\beta_0^2} \left(\log^2 L - \log L - 1\right) + \frac{\beta_2}{\beta_0}\right)\right). \tag{3}$$

The coefficients of the QCD beta function are given in the $\overline{\text{MS}}$ scheme as

$$\beta_0 = \frac{11 C_A - 4 T_R N_F}{6},$$
$$\beta_1 = \frac{17 C_A^2 - 10 C_A T_R N_F - 6 C_F T_R N_F}{6}, \tag{4}$$
$$\beta_2 = \frac{2857 C_A^3 + 108 C_F^2 T_R N_F - 1230 C_F C_A T_R N_F - 2830 C_A^2 T_R N_F + 264 C_F T_R^2 N_F^2 + 316 C_A T_R^2 N_F^2}{432}.$$

For general scale choices $\mu$, the expression eq. (1) therefore generalises to

$$
\frac{1}{\Gamma_{jj}^{(n)}(s,\mu^2)} \frac{\mathrm{d}\Gamma}{\mathrm{d}y}(s,\mu^2) = \left(\frac{\alpha_{\mathrm{s}}(\mu)}{2\pi}\right) \frac{\mathrm{d}\bar{A}}{\mathrm{d}y}
$$
$$
+ \left(\frac{\alpha_{\mathrm{s}}(\mu)}{2\pi}\right)^2 \left[\frac{\mathrm{d}\bar{B}}{\mathrm{d}y} + \beta_0 \log\left(\frac{\mu^2}{s}\right)\frac{\mathrm{d}\bar{A}}{\mathrm{d}y}\right]
$$
$$
+ \left(\frac{\alpha_{\mathrm{s}}(\mu)}{2\pi}\right)^3 \left[\frac{\mathrm{d}\bar{C}}{\mathrm{d}y} + 2\beta_0 \log\left(\frac{\mu^2}{s}\right)\frac{\mathrm{d}\bar{B}}{\mathrm{d}y}\right.
$$
$$
\left. + \left(\beta_0^2 \log^2\left(\frac{\mu^2}{s}\right) + \beta_1 \log\left(\frac{\mu^2}{s}\right)\right)\frac{\mathrm{d}\bar{A}}{\mathrm{d}y}\right] + \mathcal{O}(\alpha_{\mathrm{s}}^4). \tag{5}
$$

Instead of calculating the perturbative coefficients $\bar{A}$, $\bar{B}$, and $\bar{C}$ directly, EERAD3 calculates the related coefficients $A$, $B$, and $C$, which are defined via

$$
\frac{1}{\Gamma_{jj}^{(0)}(s)} \frac{\mathrm{d}\Gamma}{\mathrm{d}y}(s) = \left(\frac{\alpha_{\mathrm{s}}(\sqrt{s})}{2\pi}\right)\frac{\mathrm{d}A}{\mathrm{d}y} + \left(\frac{\alpha_{\mathrm{s}}(\sqrt{s})}{2\pi}\right)^2 \frac{\mathrm{d}B}{\mathrm{d}y} + \left(\frac{\alpha_{\mathrm{s}}(\sqrt{s})}{2\pi}\right)^3 \frac{\mathrm{d}C}{\mathrm{d}y} + \mathcal{O}(\alpha_{\mathrm{s}}^4). \tag{6}
$$

Note the normalisation to the Born-level decay width $\Gamma_{jj}^{(0)}$. At order $\alpha_{\mathrm{s}}^n$, the barred and un-barred coefficients are related by

$$
\bar{A} = \frac{\Gamma_{jj}^{(0)}}{\Gamma_{jj}^{(n)}}A = \frac{1}{K_{jj}^{(n)}}A, \qquad \bar{B} = \frac{\Gamma_{jj}^{(0)}}{\Gamma_{jj}^{(n)}}B = \frac{1}{K_{jj}^{(n)}}B, \qquad \bar{C} = \frac{\Gamma_{jj}^{(0)}}{\Gamma_{jj}^{(n)}}C = \frac{1}{K_{jj}^{(n)}}C. \tag{7}
$$

In this context, $K^{(n)}$ denotes the $n$-th order inclusive $K$-factor of the relevant process,

$$
K_{jj}^{(n)} = 1 + V_{jj}^{(1)}\left(\frac{\alpha_{\mathrm{s}}}{2\pi}\right) + V_{jj}^{(2)}\left(\frac{\alpha_{\mathrm{s}}}{2\pi}\right)^2 + \dots. \tag{8}
$$

If desired, the ratio $1/K^{(n)}$ can be expanded up to the relevant perturbative order to yield

$$
\bar{A} = A, \qquad \bar{B} = B - V_{jj}^{(1)}A, \qquad \bar{C} = C - V_{jj}^{(1)}B + \left(\left(V_{jj}^{(1)}\right)^2 - V_{jj}^{(2)}\right)A. \tag{9}
$$

In case of off-shell photon or $Z$-boson decays, the $\gamma^*/Z \to q\bar{q}$ decay width $\Gamma$ in eq. (1) is replaced by the $e^+e^- \to q\bar{q}$ cross section $\sigma_{q\bar{q}}$. At Born level, it is given by

$$
\sigma_{q\bar{q}}^{(0)}(s,\mu^2) = \frac{4\pi\alpha}{3s}N_{\mathrm{C}}\sum_q e_q^2, \tag{10}
$$

with the electromagnetic coupling constant

$$
\alpha = M_W^2\left(1 - \frac{M_W^2}{M_Z^2}\right)\frac{\sqrt{2}G_{\mathrm{F}}}{\pi} \tag{11}
$$

and the centre-of-mass energy $s$. Up to second order in QCD, the perturbative expansion of $\sigma_{q\bar{q}}$ is given by

$$
\sigma_{q\bar{q}}^{(2)}(s,\mu^2) = \sigma_{q\bar{q}}^{(0)}(s,\mu^2)K_{q\bar{q}}^{(2)}(s,\mu^2)
$$
$$
= \sigma_{q\bar{q}}^{(0)}(s,\mu^2)\left[1 + V_{q\bar{q}}^{(1)}(s,\mu^2)\left(\frac{\alpha_{\mathrm{s}}(\mu)}{2\pi}\right) + V_{q\bar{q}}^{(2)}(s,\mu^2)\left(\frac{\alpha_{\mathrm{s}}(\mu)}{2\pi}\right)^2\right]. \tag{12}
$$

Here, the NLO and NNLO corrections to the inclusive $K$-factor $V_{q\bar{q}}^{(1)}$ and $V_{q\bar{q}}^{(2)}$ are given by [47]:

$$V_{q\bar{q}}^{(1)}(s, \mu^2) = \frac{3}{2} C_{\rm F}, \tag{13}$$

$$V_{q\bar{q}}^{(2)}(s, \mu^2) = -\frac{3}{8} C_{\rm F}^2 + \left( \frac{123}{8} - 11\zeta_3 - \frac{11}{4} \log\left( \frac{\mu^2}{s} \right) \right) C_{\rm F} C_{\rm A}$$
$$+ \left( -\frac{11}{2} + 4\zeta_3 + \log\left( \frac{\mu^2}{s} \right) \right) C_{\rm F} T_{\rm R} N_{\rm F}. \tag{14}$$

For Higgs-boson decays to a massless $b$-quark pair or two gluons, the inclusive decay widths are given at Born level by

$$\Gamma_{b\bar{b}}^{(0)}(s, \mu^2) = \frac{y_b^2(\mu) N_{\rm C} \sqrt{s}}{8\pi}, \quad \Gamma_{gg}^{(0)}(s, \mu^2) = \frac{\lambda_0^2(\mu)(N_{\rm C}^2 - 1)\sqrt{s}^3}{64\pi}, \tag{15}$$

where $\sqrt{s}$ denotes the mass of the decaying colour singlet. The renormalised Born-level coupling constants are given in terms of the Fermi constant $G_{\rm F}$ as

$$y_b^2(\mu) = m_b^2(\mu) \sqrt{2} G_{\rm F}, \quad \text{and} \quad \lambda_0^2(\mu) = \frac{\alpha_{\rm s}^2(\mu)}{9\pi^2} \sqrt{2} G_{\rm F}, \tag{16}$$

It should be pointed out that, different to the $\gamma^*/Z \to q\bar{q}$ decay, the decay widths $\Gamma_{b\bar{b}}$ and $\Gamma_{gg}$ depend on the renormalisation scale $\mu$ already at Born level. This dependence enters through the renormalisation of the $b$-quark Yukawa coupling $y_b$ in the $H \to b\bar{b}$ case and the renormalisation of the strong coupling $\alpha_{\rm s}$ in the $H \to gg$ case, where it enters at Born level through the effective $Hgg$ coupling due to the integrated top-quark loop. The first-order QCD corrections to the decay widths

$$\Gamma_{H \to b\bar{b}}^{(1)}(s, \mu^2) = \Gamma_{H \to b\bar{b}}^{(0)}(s, \mu^2) \left[ 1 + V_{b\bar{b}}^{(1)}(s, \mu^2) \left( \frac{\alpha_{\rm s}(\mu)}{2\pi} \right) \right] \tag{17}$$

$$\Gamma_{H \to gg}^{(1)}(s, \mu^2) = \Gamma_{H \to gg}^{(0)}(s, \mu^2) \left[ 1 + V_{gg}^{(1)}(s, \mu^2) \left( \frac{\alpha_{\rm s}(\mu)}{2\pi} \right) \right]. \tag{18}$$

where the NLO corrections to $K_{b\bar{b}}$ and $K_{gg}$ are given by [48–59]

$$V_{b\bar{b}}^{(1)}(s, \mu^2) = \frac{17}{2} C_{\rm F} + 3 C_{\rm F} \log\left( \frac{\mu^2}{s} \right), \tag{19}$$

$$V_{gg}^{(1)}(s, \mu^2) = \frac{95}{6} C_{\rm A} - \frac{14}{3} T_{\rm R} N_{\rm F} + 2\beta_0 \log\left( \frac{\mu^2}{s} \right). \tag{20}$$

# 3 Available processes

An overview of the currently available processes and their perturbative order is given in table 1.

## 3.1 $\gamma^*/Z \to q\bar{q}$

The following processes initiated by a $\gamma^*/Z \to q\bar{q}$ decay are available:

- $\gamma^*/Z \to 3j$ at NNLO

- $\gamma^*/Z \to 4j$ at NLO

| ID | Process | $3j$ | $4j$ | $5j$ |
|----|---------|------|------|------|
| 1  | $Z \to q\bar{q}$ | NNLO | NLO | LO |
| 21 | $H \to q\bar{q}$ | NLO | NLO | LO |
| 22 | $H \to gg$ (HTL) | NLO | NLO | LO |

Table 1: Available processes and their respective perturbative order in QCD as of EERAD3 version 2.0.

- $\gamma^*/Z \to 5j$ at LO

In all cases, light quarks (including the $b$-quark) are treated as massless.

Tree-level matrix elements with up to four partons and three-parton one-loop amplitudes are constructed from real-radiation antenna functions [16, 60]. Five-parton tree-level matrix elements are calculated explicitly using FORM [61], four-parton one-loop amplitudes are taken from the calculation in [62] and three-parton two-loop amplitudes from [63, 64]. The special functions in the two-loop amplitudes are evaluated using the HPLOG [65] and TDHPL [66] routines, which are included in the EERAD3 distribution.

## 3.2   $H \to b\bar{b}$

The following processes initiated by the $H \to b\bar{b}$ decay are available:

- $H \to 3j$ at NLO

- $H \to 4j$ at NLO

- $H \to 5j$ at LO

In all cases, $b$-quarks are treated as massless with non-vanishing Yukawa coupling.

Tree-level matrix elements with up to four partons are calculated explicitly using FORM [61]. Tree-level five-parton matrix elements are taken from the N$^3$LO $H \to b\bar{b}$ and NNLO $H \to b\bar{b}j$ calculation in [67, 68], in turn calculated using BCFW recursion relations [69]. The three-parton one-loop matrix element are adapted from the calculation in [70, 71]. Four-parton one-loop amplitudes are adapted from the calculation in [67, 68], where they have been derived analytically by use of the generalised unitarity approach [72], using quadruple cuts for box coefficients [73], triple cuts for triangle coefficients [74], double cuts for bubble coefficients [75], and $d$-dimensional unitarity techniques for the rational pieces [76, 77].

It should be noted that, although not implemented as such, predictions for $H \to c\bar{c}$ can be obtained trivially by changing the mass value input parameter that is used to compute the Yukawa coupling.

## 3.3   $H \to gg$

The following processes initiated by the $H \to gg$ decay are available:

- $H \to 3j$ at NLO

- $H \to 4j$ at NLO

- $H \to 5j$ at LO

In all cases, top-quarks are treated as infinitely heavy and the top-quark loop is integrated out, so that the Higgs couples directly to gluons via an effective coupling.

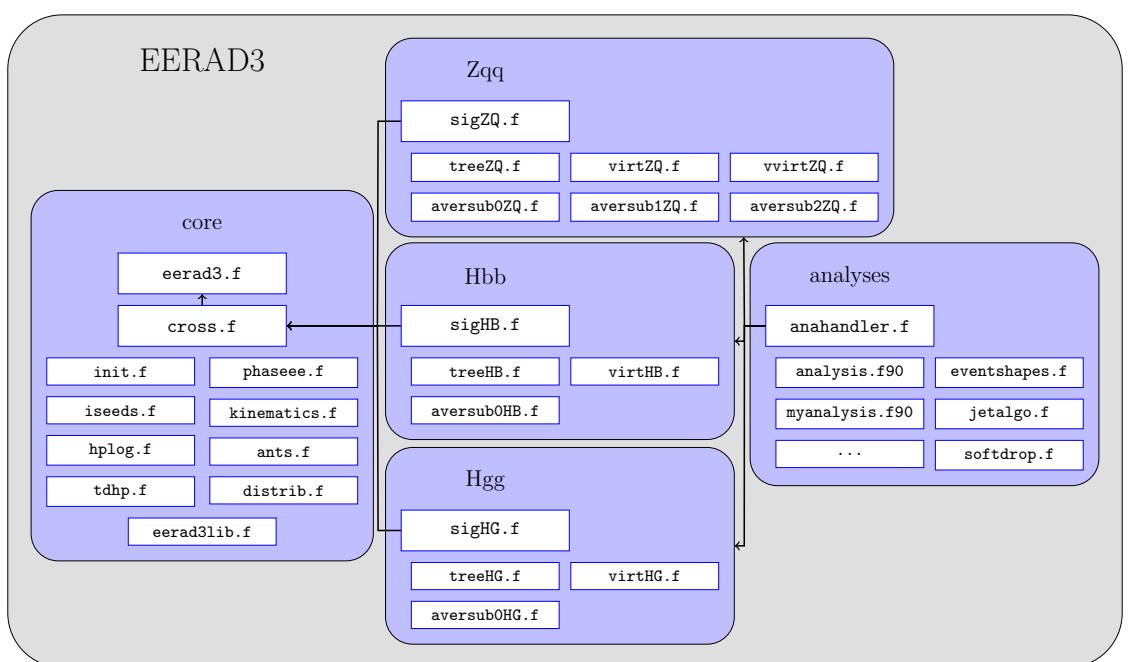

Figure 1: Structure of EERAD3 v2.

Tree-level matrix elements with up to four partons are constructed directly from gluon-gluon antenna functions [78]. Five-parton tree-level amplitudes are obtained by crossing the ones used in the $pp \to Hj$ NNLO calculation in [79, 80], which are based on the results presented in [81–84]. Three-parton one-loop amplitudes are adapted from the implementation in [85, 86], in turn based on the results given in [87]. Four-parton one-loop amplitudes are obtained by crossing the ones used in the $pp \to Hj$ NNLO calculation in [79, 80], which are based on the results presented in [81, 84, 88–94].

At NLO, the differential $H \to gg$ decay rate further receives contributions from the $\mathcal{O}(\alpha_{\mathrm{s}})$ expansion of the top-quark matching coefficient [57, 95, 96]. These are implemented as a finite contribution to the virtual correction,

$$\mathrm{d}\Gamma^{\mathrm{V}} \to \mathrm{d}\Gamma^{\mathrm{V}} + \left(\frac{\alpha_{\mathrm{s}}}{2\pi}\right)\frac{11}{3}N_{\mathrm{C}}\mathrm{d}\Gamma^{\mathrm{B}}. \tag{21}$$

## 4  Structure of the program

Schematically, the structure and content of EERAD3 v2 is sketched in fig. 1. The main program `eerad3.f` as well as all process-independent files except for analyses are contained in the `core/` folder. The calculation of the differential cross section is handled in the `cross.f` file, which interfaces the subroutines and functions in `sigZQ.f`, `sigHB.f`, or `sigHG.f`, depending on the process. Several other files are contained in the `core/` folder, which are used globally in EERAD3. These are:

`eerad3lib.f` library with special functions and auxiliary subroutines

`phaseee.f` phase-space generation subroutines

`kinematics.f` single-unresolved and double-unresolved antenna kinematics from [97]

`ants.f` real, real-virtual, and double-real antenna functions from [16]

`distrib.f` subroutine to optimise phase-space integration

`hplog.f` one-dimensional harmonic polylogarithms from [65]

`tdhpl.f` two-dimensional harmonic polylogarithms from [66]

For each process, the `sig*.f` file contains all subroutines relevant to the calculation of the three-particle, four-particle, and five-particle cross sections, which are called by the `cross()` subroutine in `src/core/cross.f`. The `aversub0*.f` file contains the single-unresolved subtraction terms. If available, the `aversub1*.f` contains the double-unresolved subtraction terms and `aversub2*.f` contains the single-unresolved one-loop subtraction terms. All relevant tree-level matrix elements with up to five partons are collected in the `tree*.f` files and the corresponding one-loop matrix elements with up to four partons are collected in the `virt*.f` files. If the process is available at NNLO, the relevant double-virtual matrix elements are collected in `vvirt*.f`. In case of the $\gamma^*/Z \to q\bar{q}$ process `treeZQ.f` only contains tree-level matrix elements with five partons and `virtZQ.f` contains one-loop matrix elements with four partons. All other matrix elements needed in `sigZQ.f` are expressed directly in terms of antenna functions.

The `analyses/` folder contains the main modules and subroutines required for the calculation of infrared-safe observables. The default analysis is contained in `analysis.f90` and makes use of the event-shape observables defined in `eventshapes.f`, the soft-drop algorithm in `softdrop.f`, and the Durham jet algorithm in `jetalgo.f`. An example of a user-defined analysis is given in `myanalysis.f90`, but is not compiled by default.

## 5 Usage

In this section, the usage of EERAD3 will be described. Section 5.1 summarises the steps needed to compile the program and section 5.2 provides a guide to running EERAD3. In section 5.3 the default analysis is summarised, including a list of observables and the binning of distributions, while section 5.4 is intended to help users write their own analysis. Section 5.5 describes the format of output histograms generated by EERAD3 and details the steps needed to calculate fixed-order distributions from the perturbative output.

### 5.1 Installation

The main EERAD3 program requires the `gfortran` compiler (version $\geq$ 9.0). Additional post-processing of histograms is performed with the `eerad3hist` python program, for which `python3` (version $\geq$ 3.0.1) is required. Running the following command in the top-level EERAD3 directory compiles the main program and copies the executable `eerad3` and the auxiliary `eerad3hist` histogram-handling program in the `bin/` folder:

```
make [-j <cores>] [ANALYSIS=<analysis>]
```

The compilation can be accelerated by providing multiple cores via the optional `-j` flag, e.g., `-j 4` if 4 cores shall be used for compilation. The `ANALYSIS` variable is optional and can be used to compile EERAD3 with a custom analysis named `<analysis>`, see section 5.4 for a guide on user-defined analyses.

### 5.2 Running EERAD3

EERAD3 is executed by running the following command in the terminal

| Setting | Explanation | Options | Default |
|---------|-------------|---------|---------|
| Process | | | |
| process | process ID | 1: $Z \rightarrow q\bar{q}$<br>21: $H \rightarrow b\bar{b}$<br>22: $H \rightarrow gg$ | - |
| njets | number of hard jets | 3, 4, 5 | - |
| channel | perturbative level | LO, NLO, NNLO,<br>V, R, VV, RV, RR | - |
| colour_layer | colour factor | 0: full colour sum<br>> 0: single colour factors | 0 |
| Integration | | | |
| warmup | iterations in<br>warmup run | $\geq 0$ | 5 |
| production | iterations in<br>production run | $\geq 0$ | 5 |
| shots | points per iteration | > 0 | - |
| Technical | | | |
| prefix | histogram directory | any string with<br>8 characters or less | results |
| ang_average | angular averaging | 0: off<br>1: on | 1 |
| y0 | technical cut off | $\geq 10^{-8}$ | $10^{-6}$ |
| Phase-space weighting | | | |
| sigma_obs | observable in<br>weighted integral | $\geq 0$ | 0 |
| moment | power of observable<br>in integrand | $\geq 0$ | 1 |

Table 2: Available settings, their options, and default values. Settings without default values have to be specified in the run card.

```
./eerad3 -i <runcard> [options]
```

Stating the run card is compulsory and its name has to be specified with the `-i` argument. The following optional arguments are available:

```
-s <seed>       random-number seed (0-9999); default: 0
-n <points>     number of points per iteration
-w <iterations> number of warmup iterations
-p <iterations> number of production iterations
```

The run card contains all information on the process of relevance, as well as cuts on observables and technical settings. Lines starting with an exclamation mark are disregarded as comments.

An example run card for the NLO corrections to $H \rightarrow 3j$ in the $H \rightarrow b\bar{b}$ decay category is given below:

```
! Process settings
process    = 21
njets      = 3
channel    = NLO
```

| Histogram | Min | Max | Bins |
|---|---|---|---|
| $\tau\,\mathrm{d}\sigma/\mathrm{d}\tau$ | 0.0 | 0.5 | 200 |
| $\mathrm{d}\sigma/\mathrm{d}\tau$ | 0.0 | 0.5 | 200 |
| $\mathrm{d}\sigma/\mathrm{d}\log\tau$ | −10 | 0 | 200 |
| $\mathrm{d}\sigma/\mathrm{d}\log_{10}\tau$ | −5 | 0 | 200 |
| $C\,\mathrm{d}\sigma/\mathrm{d}C$ | 0.0 | 1.0 | 400 |
| $\mathrm{d}\sigma/\mathrm{d}C$ | 0.0 | 1.0 | 400 |
| $\mathrm{d}\sigma/\mathrm{d}\log C$ | −10 | 0 | 200 |
| $\mathrm{d}\sigma/\mathrm{d}\log_{10}C$ | −5 | 0 | 200 |
| $\rho_{\mathrm{H}}\,\mathrm{d}\sigma/\mathrm{d}\rho_{\mathrm{H}}$ | 0.0 | 0.5 | 200 |
| $\mathrm{d}\sigma/\mathrm{d}\rho_{\mathrm{H}}$ | 0.0 | 0.5 | 200 |
| $\mathrm{d}\sigma/\mathrm{d}\log\rho_{\mathrm{H}}$ | −10 | 0 | 200 |
| $\mathrm{d}\sigma/\mathrm{d}\log_{\rho_{\mathrm{H}}}$ | −5 | 0 | 200 |
| $B_{\mathrm{W}}\,\mathrm{d}\sigma/\mathrm{d}B_{\mathrm{W}}$ | 0.0 | 0.5 | 200 |
| $\mathrm{d}\sigma/\mathrm{d}B_{\mathrm{W}}$ | 0.0 | 0.5 | 200 |
| $\mathrm{d}\sigma/\mathrm{d}\log B_{\mathrm{W}}$ | −10 | 0 | 200 |
| $\mathrm{d}\sigma/\mathrm{d}\log_{B_{\mathrm{W}}}$ | −5 | 0 | 200 |
| $B_{\mathrm{T}}\,\mathrm{d}\sigma/\mathrm{d}B_{\mathrm{T}}$ | 0.0 | 0.5 | 200 |
| $\mathrm{d}\sigma/\mathrm{d}B_{\mathrm{T}}$ | 0.0 | 0.5 | 200 |
| $\mathrm{d}\sigma/\mathrm{d}\log B_{\mathrm{T}}$ | −10 | 0 | 200 |
| $\mathrm{d}\sigma/\mathrm{d}\log_{B_{\mathrm{T}}}$ | −5 | 0 | 200 |
| $\mathrm{d}\sigma/\mathrm{d}\log FC_{x}$ | −10 | 0 | 200 |
| $\tau^{\mathrm{sd}}\,\mathrm{d}\sigma/\mathrm{d}\tau^{\mathrm{sd}}$ | 0.0 | 0.5 | 200 |
| $\mathrm{d}\sigma/\mathrm{d}\log\tau^{\mathrm{sd}}$ | −10 | 0 | 200 |
| $\mathrm{d}\sigma/\mathrm{d}\log_{10}\tau^{\mathrm{sd}}$ | −5 | 0 | 200 |

Table 3: Binnings of three-jet event-shape histograms implemented in the default analysis.

```
! Technical settings.
y0         = 1d-8

! Observables.
cut        = 1d-5
sigma_obs  = 0
moment     = 1

! Vegas settings.
warmup     = 5
production = 5
shots      = 1M
```

In this example, a single run with 5 iterations in the warm-up and production phase and 1M phase-space points is performed. The technical cut-off to prevent kinematic invariants to go deeply into infrared singular limits is set to $10^{-8}$, while the observable cut is set to $10^{-5}$.

## 5.3 Default analysis

The following observables are implemented in the default analysis:

| Histogram | Min | Max | Bins |
|---|---|---|---|
| $T_{\text{Minor}} \, \mathrm{d}\sigma/\mathrm{d}T_{\text{Minor}}$ | 0.0 | 0.5 | 200 |
| $\mathrm{d}\sigma/\mathrm{d}T_{\text{Minor}}$ | 0.0 | 0.5 | 200 |
| $\mathrm{d}\sigma/\mathrm{d}\log T_{\text{Minor}}$ | $-10$ | 0 | 200 |
| $\mathrm{d}\sigma/\mathrm{d}\log_{10} T_{\text{Minor}}$ | $-8$ | 0 | 200 |
| $D \, \mathrm{d}\sigma/\mathrm{d}D$ | 0.0 | 1.0 | 400 |
| $\mathrm{d}\sigma/\mathrm{d}D$ | 0.0 | 1.0 | 400 |
| $\mathrm{d}\sigma/\mathrm{d}\log D$ | $-10$ | 0 | 200 |
| $\mathrm{d}\sigma/\mathrm{d}\log_{10} D$ | $-8$ | 0 | 200 |
| $\rho_{\text{L}} \, \mathrm{d}\sigma/\mathrm{d}\rho_{\text{L}}$ | 0.0 | 0.2 | 200 |
| $\mathrm{d}\sigma/\mathrm{d}\rho_{\text{L}}$ | 0.0 | 0.2 | 200 |
| $\mathrm{d}\sigma/\mathrm{d}\log \rho_{\text{L}}$ | $-10$ | 0 | 200 |
| $\mathrm{d}\sigma/\mathrm{d}\log_{10} \rho_{\text{L}}$ | $-8$ | 0 | 200 |
| $B_{\text{N}} \, \mathrm{d}\sigma/\mathrm{d}B_{\text{N}}$ | 0.0 | 0.2 | 200 |
| $\mathrm{d}\sigma/\mathrm{d}B_{\text{N}}$ | 0.0 | 0.2 | 200 |
| $\mathrm{d}\sigma/\mathrm{d}\log B_{\text{W}}$ | $-10$ | 0 | 200 |
| $\mathrm{d}\sigma/\mathrm{d}\log_{10} B_{\text{W}}$ | $-8$ | 0 | 200 |

Table 4: Binnings of four-jet event-shape histograms implemented in the default analysis.

- Durham jet rates $R_3^{\text{D}}$, $R_4^{\text{D}}$, and $R_5^{\text{D}}$ [98–102]

- Durham jet-resolution scales $y_{23}^{\text{D}}$, $y_{34}^{\text{D}}$, $y_{45}^{\text{D}}$ [98–102]

- Jet broadenings $B_{\text{T}}$, $B_{\text{W}}$, $B_{\text{N}}$ [103, 104]

- Jet Masses $\rho_{\text{H}}$, $\rho_{\text{L}}$ [105]

- Thrust $\tau = 1 - T$, $T_{\text{Minor}}$ [106, 107]

- $C$- and $D$-parameter [32, 108]

- $FC_x$ variables for $x = 0, 0.5, 1, 1.5$ [109]

- soft-drop thrust $\tau_{\text{sd}}$ for $z_{\text{cut}} = 0.1$ and $\beta = 0, 1, 2$ [110]

The binnings of for three-jet event-shape histograms are summarised in table 3, for four-jet event-shape histograms in table 4, and for jet-rate and jet-resolution histograms in table 5. Additional observables can be implemented in a custom analysis, see section 5.4.

## 5.4 Custom analyses

The default analysis already contains a large number of widely used event-shape observables. Nevertheless, EERAD3 offers the possibility to write custom analyses, e.g., to limit the set of observables in order to decrease the run time or to implement user-defined observables. Analyses must be written as `fortran` modules. A minimal example of a user analysis can be found in `myanalysis.f90` in the `src/analyses/` folder.

A user-defined analysis is expected to contain the following four subroutines:

- `initanalysis` to initialise the analysis, set cuts, and book histograms

- `ecuts_ana` to calculate observables and apply event-selection cuts

| Histogram | Min | Max | Bins |
|---|---|---|---|
| $R_3^{\mathrm{D}}(\log y_{\mathrm{cut}})$ | $-10$ | $0$ | $200$ |
| $R_4^{\mathrm{D}}(\log y_{\mathrm{cut}})$ | $-10$ | $0$ | $200$ |
| $R_5^{\mathrm{D}}(\log y_{\mathrm{cut}})$ | $-10$ | $0$ | $200$ |
| $y_{23}^{\mathrm{D}}\mathrm{d}\sigma/\mathrm{d}\log y_{23}^{\mathrm{D}}$ | $0$ | $1$ | $400$ |
| $\mathrm{d}\sigma/\mathrm{d}\log y_{23}^{\mathrm{D}}$ | $-10$ | $0$ | $200$ |
| $\mathrm{d}\sigma/\mathrm{d}\log_{10} y_{23}^{\mathrm{D}}$ | $-5$ | $0$ | $200$ |
| $y_{34}^{\mathrm{D}}\mathrm{d}\sigma/\mathrm{d}\log y_{34}^{\mathrm{D}}$ | $0$ | $1$ | $400$ |
| $\mathrm{d}\sigma/\mathrm{d}\log y_{34}^{\mathrm{D}}$ | $-10$ | $0$ | $200$ |
| $\mathrm{d}\sigma/\mathrm{d}\log_{10} y_{34}^{\mathrm{D}}$ | $-5$ | $0$ | $200$ |
| $y_{45}^{\mathrm{D}}\mathrm{d}\sigma/\mathrm{d}\log y_{45}^{\mathrm{D}}$ | $0$ | $1$ | $400$ |
| $\mathrm{d}\sigma/\mathrm{d}\log y_{45}^{\mathrm{D}}$ | $-10$ | $0$ | $200$ |
| $\mathrm{d}\sigma/\mathrm{d}\log_{10} y_{45}^{\mathrm{D}}$ | $-5$ | $0$ | $200$ |

Table 5: Binnings of jet-rate and jet-resolution histograms implemented in the default analysis.

- `fillhists` to fill histograms with observable values calculated in `ecuts`

- `getvar` to apply weights to the phase-space generation (can be unity)

The module can be used to store information on observables, cuts, and histograms.

```fortran
module analysis_mod
  implicit none

  ! Observables.
  real(8) :: obs1
  ! Cuts.
  real(8) :: cut
  ! Histogram IDs.
  integer :: iObs1

  ! Common blocks - do not touch!
  integer :: iaver,imom,idist,iang,idebug
  integer :: iproc,nloop,icol,njets,ichan
  common/intech/iaver,imom,idist,iang,idebug
  common/inphys/iproc,nloop,icol,njets,ichan

contains
  ...
end module analysis_mod
```

The `initanalysis` subroutine is used to initialise the analysis and book all histograms of interest.

```fortran
subroutine initanalysis()
  implicit none
  integer, external :: bookhist

  ! Read cuts.
  call readparm('cut', cut, 1d-5)
```

```
  ! Book histograms.
  iHist1 = bookhist('hist1', 0d0, 0.5d0, 200)

  ! Print histogram information.
  call printhistdata()

end subroutine initanalysis
```

Event-selection cuts can either be hard-coded in the analysis or read in from the run card. For the latter, the `readparm` subroutine can be used, which tries to find a setting called `sname` in the runcard and sets `variable` to the corresponding value if found or to `defaultvalue` if not.

```
subroutine readparm(sname, variable, defaultvalue)
```

Histograms are booked via the external function `bookhist`, which initialises a new histogram with name `name` and `nbin` bins with lower edge `bmin` and upper edge `bmax`.

```
integer function bookhist(hname, bmin, bmax, nbin)
```

It returns a unique integer ID for each new histogram.

In the `ecuts_ana` subroutine all observables of interest are calculated and all cuts are implemented. It takes three arguments to communicate the number of particles in the current event (`npar`), the weight for the cross section (`var`), and whether the event passes the event-selection cuts (`ipass`):

```
subroutine ecuts_ana(npar, var, ipass)
  implicit none
  integer, intent(in)    :: npar
  integer, intent(inout) :: ipass
  real(8), intent(inout) :: var

  ! Calculate observables.
  call getObs1(obs1,npar)

  ! Set variable for integration.
  call getvar(var)

  !  Apply cuts.
  if (obs1.gt.obs1cut)then
    ipass=1
  endif

end subroutine ecuts_ana
```

The `fillhists` subroutine is used to fill histograms. In order to avoid infrared divergences in the histogram when employing multi-variable cuts, it is advised to check the respective cut of each observable before adding the weight.

```
subroutine fillhists(wgt)
  implicit none
  real(8), intent(in)    :: wgt

  ! Divide out moment from event weight.
```

```
  call getvar(var)
  wt = wgt/var

  ! Fill histograms.
  if (obs1.gt.obs1)then
    call histoa(iHist1, var1, wt)
  endif

end subroutine fillhists
```

There are two ways to fill a given histogram. Histograms can be referenced by name in the `fillhist` subroutine:

```
subroutine fillhist(hname, val, wgt)
```

Alternatively, histograms can be referenced by ID via the `histoa` subroutine:

```
subroutine histoa(id, val, wgt)
```

While it is in principle supported to reference histograms by name, it is strongly advised to reference histograms by their ID in order to keep execution times minimal.

## 5.5   Output histogram files

EERAD3 produces histograms containing perturbative contributions to the coefficients $A$, $B$, and $C$ in eq. (6), corresponding to the LO, NLO, and NNLO corrections, depending on the selected jet multiplicity. The histogram files are saved in a folder with name as defined by the `prefix` setting (`results/` by default), see section 5.2. The naming convention of the files is as follows:

```
    <process>.<njets>j.<seed>.<order>.<icol>.<observable>.dat
```

Here, `<njets>` specifies the number of jets (3, 4, or 5), `<order>` refers to the perturbative correction (LO, NLO, NNLO, V, R, VV, RV, or RR), `<icol>` to the colour layer, and `<process>` refers to the following process identifiers

- Zqq: $Z \to q\bar{q}$

- Hbb: $H \to q\bar{q}$

- Hgg: $H \to gg$

An example file name for the histogram containing the real correction to the NLO coefficient $B$ for the histogram $1/\Gamma^{(0)} \, d\Gamma/d \log(\tau)$ in $H \to gg$ decays with seed 0 is

```
    Hgg.3j.0000.R.0.LogT.dat
```

Each file contains a single histogram in the format

```
    <xlow>  <xhigh>  <weight>  <error>  <count>
```

Here, `<xlow>` represents the lower bin edge, `<xhigh>` the upper bin edge, `<weight>` the sum of weights in this bin (not normalised to the bin width), `<error>` the square root of the sum of squared weights in this bin, and `<count>` the bin count.

## 5.6 EERAD3hist

The `eerad3hist` program, placed in the `bin/` folder upon compilation, allows to `merge` histograms from statistically independent runs, `combine` histograms into the corresponding perturbative coefficients $A$, $B$, and $C$ in eq. (5), and calculate physical distributions according to eq. (5). The usage of `eerad3hist` will be described in the following.

### 5.6.1 Merging histograms from indpendent runs

Histograms from statistically independent runs can be merged using the `merge` command. It can be run as follows:

```
./eerad3hist merge [-o <outdir>] [-t <tag>] <directory>
```

Here, `<tag>` defines a string that replaces the `<seed>` part of the histogram name after merging. Given a path `<directory>`, the `merge` script automatically merges all histogram files that differ only by a random-number seed in the folder `<directory>`. Uncertainties are accumulated as bin-wise weighted means. Unless specified otherwise via the `-o <outdir>` argument, the merged files are placed in a directory called `merged/`. The output of the `merge` command can be used as input to the `combine` command.

### 5.6.2 Combining histograms into perturbative coefficients

To combine separate contributions to a single perturbative coefficient, i.e., to combine V and R at NLO or VV, RV, and RR at NNLO, the `combine` command can be used. The usage is as follows:

```
./eerad3hist combine [-o <outdir>] <directory>
```

Given the path `<directory>`, the `combine` script automatically generates

- the `*.LO.*.dat` files if the `*.LO.*.dat` files are present in `<directory>` (this is just copied)

- the `*.NLO.*.dat` files if the `*.V.*.dat` and `*.R.*.dat` files are present in `<directory>`

- the `*.NNLO.*.dat` files if the `*.VV.*.dat`, `*.RV.*.dat`, and `*.RR.*.dat` files are present in `<directory>`

The generation of the `*.NLO.*.dat` and `*.NNLO.*.dat` files proceeds per process, jet multiplicity, seed, and observable. Unless specified otherwise via the `-o <outdir>` argument, the combined files are placed in a directory called `combined/`. In addition to the combined histogram files, the `combine` command also generates a template input file called `makedist.input` for the `makedist` command described in the next subsection.

### 5.6.3 Making distributions from histograms

The EERAD3 main program generates histograms of (contributions to) the perturbative coefficients $A$, $B$, and $C$ as in eq. (6). Histograms of physically meaningful distributions according to eq. (5) can be generated using the `makedist` command of `eerad3hist`. Specifically, `makedist` generates distributions normalised to the two-particle decay width $\Gamma_{jj}^{(n)}$ at order $n$, where $n = 0$ at LO, $n = 1$ at NLO, and $n = 2$ at NNLO. The `makedist` command works as follows:

```
./eerad3hist makedist [-o <outdir>] [-f <format>] [-E] [-B] <input>
```

Here, the command file `<input>` contains the process number, the number of hard jets, the centre-of-mass energy, and values of certain Standard-Model parameters. The latter should be given in the $G_\mu$-scheme, i.e., electroweak parameters are derived from the set $(G_F, m_W, m_Z)$. The command file therefore has to contain the following settings and replaces them by default values in case they are absent:

```
process = 1 | 21 | 22
sqrts   = 91.2
njets   = 3
GF      = 1.1664e-5
aS[MZ]  = 0.118
```

For the $\gamma^*/Z \to q\bar{q}$ process, the $\overline{\text{MS}}$ values of the $Z$- and $W$-mass should be specified, from which the (fixed) electromagnetic coupling is calculated:

```
MASS[W] = 80.385
MASS[Z] = 91.2
```

In case of $H \to b\bar{b}$ decays, the $\overline{\text{MS}}$ mass of the $b$-quark at $m_Z$ should be specified, which enters the $Hbb$ Yukawa coupling:

```
MASS[b] = 4.18
```

For $H \to gg$ decays, the $\overline{\text{MS}}$ $t$-quark mass at $m_Z$ should be specified, which enters the second-order correction of the $Hgg$ matching coefficient:

```
MASS[t] = 163.136
```

The running of the quark masses to the scale $\mu$ is performed at four-loop order using the results of [111]. The values of the $b$- and $t$-quark mass have no effect for the $Z \to q\bar{q}$ process. Likewise, the values of the $Z$- and $W$-boson masses have no effect on the $H \to b\bar{b}$ and $H \to gg$ processes.

Two different options are available to normalise the distributions to the respective order of the Born-level decay width according to eq. (5). By default, the distributions are simply scaled by the decay width at the respective order, as given in eq. (7). Formally, this procedure includes contributions at higher orders in the strong coupling. Upon specifying the command

```
--expand-norm (-E)
```

the normalisation is expanded to the respective order as in eq. (9). In this case, no higher-order contributions are included.

For $H \to b\bar{b}$ and $H \to gg$ decays, the distributions can be weighted by the respective branching ratio. This behaviour is disabled by default and can be enabled using the flag

```
--with-branching-ratio (-B)
```

At $\mu = M_H = 125.09$ GeV, the relative branching ratios at NLO are, cf. [112],

$$\text{BR}^{(1)}_{H \to b\bar{b}}(M_H) = 0.8915, \quad \text{BR}^{(1)}_{H \to gg}(M_H) = 0.1085. \tag{22}$$

For each histogram, the `makedist.input` card requires an entry in the `HISTOGRAMS` section in the following format:

```
HISTOGRAMS
<name1>  <LO file1>  [<NLO file1>]  [<NNLO file1>]
<name2>  <LO file2>  [<NLO file2>]  [<NNLO file2>]
...
END HISTOGRAMS
```

Here, the `<NLO file>` and `<NNLO file>` entries are optional. If only the name `<LO file>` is given, the histogram `<histogram>` is generated at LO; similarly, if only `<LO file>` and `<NLO file>` are given, the histogram `<histogram>` is generated at NLO.

For each observable and each perturbative order, the `makedist` command generates a separate histogram in a native plain-text format, called `<name>.LO.dat`, `<name>.NLO.dat`, and `<name>.NNLO.dat`. In the native plain-text format, histograms are written as

```
<xlow>  <xhigh>  <sig>  <mcerror>  <vardown>  <varup>
```

where `<xlow>` and `<xhigh>` denotes the bin edges, `<sig>` the sum of weights in the bin (divided by the bin width), `<mcerror>` the statistical error estimate, and `<vardown>` and `<varup>` the up and down scale variations. These can be plotted using plotting tools such as `pyplot` or `gnuplot`. For simple comparisons to other event-generation frameworks, `eerad3hist` also supports the YODA format [113], available with the `-f yoda` option. In this case, histograms of different observables are combined into a single file for each perturbative order called `LO.yoda`, `NLO.yoda`, and `NNLO.yoda`. These can be plotted using the plotting scripts shipped with RIVET [114, 115]. Unless otherwise specified via the `-o <outdir>` option, either of the histogram files are placed in a folder called `hist/`.

## 6 Summary

We have presented a major update of the EERAD3 parton-level event-generation framework for fixed-order QCD calculations in hadronic colour-singlet resonance decays. In EERAD3 v2, the $\gamma^*/Z \to 3j$ process can be calculated up to NNLO, while both the $H \to 3j$ and $H \to 4j$ processes include NLO corrections in the Yukawa-induced and $Hgg$-induced decay categories. Compared to version 1, EERAD3 v2 has seen a substantial structural overhaul and contains many new event-shape observables. Specifically, the analysis framework was rewritten to facilitate a simple implementation of user-defined observables. Also, modern histogram formats are supported via a `python`-based post-processing tool. The new version and future updates will be made publicly available at gitlab.com/eerad-team/releases.

## Acknowledgements

The authors would like to thank Ciaran Williams for his contributions to the Higgs-decay matrix elements used in EERAD3 and Simone Caletti and Francesco Merlotti for testing the pre-release versions of the code and for comments on the manuscript. NG gretafeully acknowledges support from the UK Science and Technology Facilities Council (STFC) under contract ST/X000745/1. AG acknowledges the support of the Swiss National Science Foundation (SNF) under contract 200021-231259. GH and BCA acknowledge the support by the Deutsche Forschungsgemeinschaft (DFG, German Research Foundation) under grant 396021762 - TRR 257.

## A Example run cards

In this appendix, we collect run cards for various setups used to obtain results in previous publications. We will use the notation

```
channel    = LO | V | R | VV | RV | RR
```

to indicate that either of the settings on the right-hand side can be used, depending on the perturbative contribution that is requested. All of the run cards presented here are made contained in the `examples/` folder of the release.

## A.1   Three-jet production in $\gamma^*/Z \to q\bar{q}$ at NNLO

A sample run card for the different perturbative contributions to the three-jet event shapes listed in section 5.3 in $Z \to q\bar{q}$ is given by:

```
! Process settings
process   = 1
njets     = 3
channel   = LO | V | R | VV | RV | RR

! Technical settings.
y0        = 1d-8

! Observables.
cut       = 1d-5
sigma_obs = 0
moment    = 1

! Vegas settings.
warmup    = 5
production = 5
shots     = 1M
```

The corresponding results have been presented in [9, 14].

## A.2   Four-jet production in $\gamma^*/Z \to q\bar{q}$ at NLO

A sample run card for the different perturbative contributions to four-jet event shapes listed in section 5.3 in $Z \to q\bar{q}$ is given by:

```
! Process settings
process   = 1
njets     = 4
channel   = LO | V | R

! Technical settings.
y0        = 1d-8

! Observables.
cut       = 1d-5
sigma_obs = 0
moment    = 1

! Vegas settings.
warmup    = 5
production = 5
shots     = 1M
```

The corresponding results have been presented in [37].

## A.3   Three-jet production in $H \to b\bar{b}$ at NLO

A sample run card for the different perturbative contributions to three-jet event shapes listed in section 5.3 in $H \to b\bar{b}$ is given by:

```
! Process settings
process    = 21
njets      = 3
channel    = LO | V | R

! Technical settings.
y0         = 1d-8

! Observables.
cut        = 1d-5
sigma_obs  = 0
moment     = 1

! Vegas settings.
warmup     = 5
production = 5
shots      = 1M
```

The corresponding results have been presented in [39].

## A.4   Four-jet production in $H \to b\bar{b}$ at NLO

A sample run card for the different perturbative contributions to four-jet event shapes listed in section 5.3 in $H \to b\bar{b}$ is given by:

```
! Process settings
process    = 21
njets      = 4
channel    = LO | V | R

! Technical settings.
y0         = 1d-8

! Observables.
cut        = 1d-5
sigma_obs  = 0
moment     = 1

! Vegas settings.
warmup     = 5
production = 5
shots      = 1M
```

The corresponding results have been presented in [44].

## A.5   Three-jet production in $H \to gg$ at NLO

A sample run card for the different perturbative contributions to three-jet event shapes listed in section 5.3 in $H \to b\bar{b}$ is given by:

```
! Process settings
process    = 22
njets      = 3
channel    = LO | V | R

! Technical settings.
y0         = 1d-8

! Observables.
cut        = 1d-5
sigma_obs  = 0
moment     = 1

! Vegas settings.
warmup     = 5
production = 5
shots      = 1M
```

The corresponding results have been presented in [39].

### A.6 Four-jet production in $H \to gg$ at NLO

A sample run card for the different perturbative contributions to four-jet event shapes listed in section 5.3 in $H \to gg$ is given by:

```
! Process settings
process    = 22
njets      = 4
channel    = LO | V | R

! Technical settings.
y0         = 1d-8

! Observables.
cut        = 1d-5
sigma_obs  = 0
moment     = 1

! Vegas settings.
warmup     = 5
production = 5
shots      = 1M
```

The corresponding results have been presented in [44].

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
