# Peer review of "EERAD3 version 2: QCD corrections in hadronic colour-singlet decays"

_SciPost Physics Codebases_

## Round 1 · Referee Report · Anonymous (Referee 1) · 2025-6-9

Report

The manuscript documents version 2 of the program EERAD3, which can be
used to calculate observables related to $\gamma^\at/Z \rightarrow \mbox{jets}$
at NNLO and Higgs decays to jets at NLO.
The manuscript is very well structured and helps the user to run the program
EERAD3.
After a minor revision, I would recommend the manuscript for publication in SciPost Physics Codebases.

Requested changes

  1. It might be worth adding a short paragraph, describing exactly the changes with respect to version 1. The information given at the end of section 1 does not seem to be inline with the information given at the end of section 6.

  2. In the caption of table 1 one might define the abbreviation "HTL", which probably stands for heavy top loop.

Recommendation

Ask for minor revision

  • validity: -
  • significance: -
  • originality: -
  • clarity: -
  • formatting: -
  • grammar: -

Author:  Christian Tobias Preuss  on 2025-08-12  [id 5721]

(in reply to Report 1 on 2025-06-09)

We would like to thank the referee for their suggestions, which we have now implemented in a revised version of the manuscript. Specifically, we have explained the abbreviation "HTL" in the caption of table 1 and we have streamlined the summary of updates between the introduction and conclusions, now giving a more concrete overview of the changes with respect to version 1.

---

## Round 1 · Referee Report · Anonymous (Referee 2) · 2025-6-25

Report

The paper documents an upgrade to eerad3, a fortran program for the evaluation of higher-order QCD corrections to event shapes and differential distribution in Higgs-boson decay, and $e^+e^-$ annihilation through a Z-boson or an off-shell photon. The software is able to provide predictions up to next-to-next-to-leading order in perturbation theory for final states of three jets. It has been used in the past for a multitude of analyses, and is a recognised tool for this type of problems. The upgrade includes technical advances and an improved interface. The usage of the software is well documented in the text. The code will remain value in the future.

Recommendation

Publish (easily meets expectations and criteria for this Journal; among top 50%)

---

## Editorial Decision

resubmitted